# People with Autism Spectrum Disorder Could Interact More Easily with a Robot than with a Human: Reasons and Limits

**DOI:** 10.3390/bs14020131

**Published:** 2024-02-12

**Authors:** Marion Dubois-Sage, Baptiste Jacquet, Frank Jamet, Jean Baratgin

**Affiliations:** 1Laboratoire Cognitions Humaine et Artificielle, RNSR 200515259U, UFR de Psychologie, Université Paris 8, 93526 Saint-Denis, France; marion.dubois-sage02@univ-paris8.fr (M.D.-S.); baptiste.jacquet@paris-reasoning.eu (B.J.); frank.jamet@paris-reasoning.eu (F.J.); 2Association P-A-R-I-S, 75005 Paris, France; 3UFR d’Éducation, CY Cergy Paris Université, 95000 Cergy-Pontoise, France

**Keywords:** autism spectrum disorder, socially assistive robot, human–robot interaction, social skills, social motivation, social cognition

## Abstract

Individuals with Autism Spectrum Disorder show deficits in communication and social interaction, as well as repetitive behaviors and restricted interests. Interacting with robots could bring benefits to this population, notably by fostering communication and social interaction. Studies even suggest that people with Autism Spectrum Disorder could interact more easily with a robot partner rather than a human partner. We will be looking at the benefits of robots and the reasons put forward to explain these results. The interest regarding robots would mainly be due to three of their characteristics: they can act as motivational tools, and they are simplified agents whose behavior is more predictable than that of a human. Nevertheless, there are still many challenges to be met in specifying the optimum conditions for using robots with individuals with Autism Spectrum Disorder.

## 1. Introduction

Autism Spectrum Disorder (ASD) is a neurodevelopmental disorder characterized by deficits in communication and social interaction as well as restricted or repetitive patterns of behavior, interests, or activities [1]. A neurobiological dysfunction is thought to be at the root of this disorder [2]. Symptoms appear early in development and have a major impact on the child’s daily life in many contexts, particularly in situations of social interaction [3].

Robotics, a currently expanding field, could be of great interest in improving support for specific populations and in particular individuals with ASD [4]. Robots can be categorized according to the different functions they perform [5]. Examples include social robots, which specialize in interacting with humans through gestures and speech, and assistance robots, which aim to help people with special needs. The use of robots has recently expanded into a new field of application: socially assistive robots, which aim specifically to foster engagement in social interactions with specific populations.

A social interaction (with a human or robotic agent) can be defined as an interaction situation in which individuals mobilize social behaviors typically used in human–human interactions. A conversation is an example of social interaction [6], but social interaction does not necessarily involve language and can also refer to any orientation towards others [7]. This notion is close to that of engagement, defined as a collaborative effort and more precisely as “the process by which two (or more) participants establish, maintain and end their perceived connection” [8] (Sidner 2002, p. 123).

Individuals with ASD often have difficulty initiating social interactions with others [9,10]. Socially assistive robots could be of great interest to people with ASD, aiming to foster the development of cognitive and social abilities [2,11], in particular by helping them to initiate more interactions [12,13,14]. Indeed, the growing number of studies showing the benefits of a robot on the social skills of individuals with ASD raises the question of whether robots enable simpler social interactions than humans. Some studies even suggest that autistic individuals would prefer to interact with a robot partner rather than a human partner, unlike Typically Developing (TD) individuals [5,15].

Several research questions will be analyzed in this review. (Q1) What benefits do individuals with ASD derive from interacting with robots? (Q2) Do individuals with ASD prefer to interact with a robot rather than a human? (Q3) If so, what reasons have been put forward in the literature to explain this phenomenon? (Q4) What are the current challenges of interacting with robots? Each of these questions will be addressed in the following sections.

## 2. Benefits of Interacting with a Robot for Individuals with ASD

According to the DSM-5 classification, the two main criteria for autism are impaired communication and social interaction on the one hand and restricted interests and repetitive behaviors on the other [1]. A review of the literature shows that robots can address both symptoms: their use with autistic people is generally aimed at improving communication skills and reducing repetitive behaviors [16]. Thus, interventions are particularly geared towards communication and social interaction difficulties but also focus on more specific behaviors, such as learning appropriate behaviors and reducing maladaptive behaviors (stereotyped behaviors and anxiety).

To determine the benefits of robots for individuals with ASD, we first conducted a search on Google Scholar using the following keywords: “autism + robot + benefits”, yielding 24,900 results. As this review is not intended to be systematic, we were particularly interested in experimental papers dealing with the progress observed in individuals with ASD following interaction with a robot (see [17] for a systematic review). We have excluded experimental papers dealing with the use of virtual robots, focusing on the use of robots in an interactive setting. In all, we selected 50 experimental studies, with publication years ranging from 2008 to 2023.

Three types of interventions can be distinguished according to the objective sought [18]: robots can promote communication and social interaction [13,14,15,19], supporting the learning of specific behaviors, such as emotion recognition [20,21], and reducing the frequency of behaviors deemed maladaptive, including repetitive behaviors and anxiety [22].

### 2.1. Fostering Communication and Social Interaction

Social assistance robots have positive effects on the social abilities of children with ASD, who generally show more social behaviors during interaction with a robot than with a human [15,23]. This benefit can be observed across a range of social behaviors impacted by ASD: eye contact, joint attention, collaborative play and activity engagement skills, touch, verbal communication, and imitation. We will list the effects of a robot for each of these behaviors (see Table 1 for a summary).

Many individuals with ASD orient less to social stimuli than TD individuals [57], with less attention paid to the eyes [58] and difficulties in initiating and maintaining eye contact [59]. An intervention based on a robotic tool with autistic children showed an increase in eye contact with a robot (with NAO [32,34]; with PROBO [33]; with KASPAR [27,35]; with QTrobot [28]; and with Tito [31]) but also towards human partners [19]. Children with autism generally looked longer at the robot than at the human [15,25,28,30] and fixed their gaze more on a robotic face than on a human face [24,26,29], which seems to indicate higher interest in a robot partner than in a human partner. In the latter study, it should be noted that autistic participants looked at the robot face as much as TD participants (in contrast to the human face, for which they showed less attention). It could be that robots can teach autistic children to establish and maintain eye contact.

Linked to difficulties with eye contact, individuals with ASD also show atypical patterns of joint attention [57]. Yet, joint attention seems to increase over time when using the NAO robot with children with ASD aged 2–4 years [42] and 6–15 years [41]. Two other studies carried out, respectively, with a CuDDler bear robot and a CommU robot also demonstrated improvement in this behavior in children aged 4–5 and 5–6 years [38,39]. Joint attention training with robots (HUMANE) is more effective than training with humans in improving response and initiation of joint attention in 6–9-year-old children with ASD [40]. In a gaze cueing paradigm, adults with ASD followed the gaze of an EDDIE robot more than that of a human [43]. Nevertheless, other studies comparing joint attention towards a robot or human partner failed to show an increase in this behavior with the robot [14,26], and sometimes instead demonstrate a decrease [36,37]. The reduction in joint attention behavior observed in these studies could be explained by a distracting effect of the robot: children would focus their attention on the robot rather than on the target object [26]. Methodological limitations, which will be discussed in detail later, may also explain these discrepancies in results. Future studies will be needed to assess the benefits of robots on joint attention.

We have observed that the social difficulties encountered by people with ASD are one of the central symptoms of autism and result in a lower tendency to initiate social interactions [10]. Yet, children with ASD engage more in collaborative play in the presence of a PROBO robot than with a human partner and increase their participation in activities that include it [45]. The presence of a COZMO robot generated more progress in interaction initiation than the same intervention without it [14]. Comparable results were obtained with a flower-shaped DAISY robot [46] and a dog-shaped AIBO robot [48]. Furthermore, interaction of autistic children (aged 6 to 8) with a KASPAR robot promoted better cooperation with a play partner [35], and, similarly, the PLEO dinosaur robot encouraged children aged 4 to 12 to interact with a third party, and to emit social behaviors [44]. The initiation of interactions is thus facilitated by working with robots [14,46], and this effect can be generalized to human–human interactions [19]. Indeed, autistic children collaborated more with an adult when they had previously interacted with a KASPAR robot [35]. In a therapeutic setting, sessions conducted in the presence of an NAO robot tend to increase autistic children’s engagement over time [47].

This increased interactivity with a robot is also found at the tactile level: autistic children are more likely to touch a robot than a human [15], and robots lead to an increase in spontaneous touch [27]. A study analyzing the spontaneous interaction of children with ASD with a robot distinguished four levels of interaction initiated by the child: exploratory interactions (touch and visual inspection); relational interactions (using another object with the robot); functional interactions (imitating the robot and dialogue); and social overtures directed towards the adult in connection with the robot (talking about the robot with the adult) [60]. The authors observed that children with ASD frequently engaged in exploratory and functional interactions with the robot spontaneously (i.e., not induced by the adult). Social openness-type interactions with the adult in connection with the robot were also observed [60].

The presence of a robot would also favor the verbal and non-verbal communication abilities of children with ASD [19,49]. Studies show a notable increase in speech production in the presence of a PLEO robot partner compared with a human partner [44], and more so with an AIBO dog robot than with a simple mechanical dog toy [48]. On the contrary, one study showed less vocalization and speech during a task performed with a robot than with a human [15], but this may be due to the short duration of the intervention (only one session of interaction). Indeed, the number of interactions performed with a KASPAR robot was positively correlated with the development of communicative abilities [51]. Interventions involving a robot significantly improve verbal communication over time (with NAO and ALICE-R50 [41]), and encourage children to ask questions spontaneously (although in this study the benefit is comparable to interventions involving only a human) [13]. In adults with ASD, job interview training with an android robot resulted in greater improvement in non-verbal communication skills (posture, gaze, voice volume, and facial expressions) than training with a human [50]. Thus, robots could help individuals with ASD to develop verbal and non-verbal communication skills.

Imitating the movements of others is also an ability that can be impacted in ASD [61,62,63]. This ability could potentially be facilitated by a robotic stimulus: autistic children imitate facial expressions of joy (smiling) from a robot more than from a human [31], and imitation of a gesture is faster when it is performed by a robotic arm rather than a human arm [53] in the robotic arm condition, the autistic children performed movements significantly faster than the TD children whereas in the human arm condition, the TD children performed movements significantly faster than the autistic children) and of better quality [56]. Training to imitate facial expressions with a robot increases performance more than the same training with a human [54]. Children with autism can thus improve their imitation skills through training with a robot, and then generalize them to interactions with humans, a result that is maintained 3 months after the intervention [52]. Nevertheless, the improvements in imitation brought about by a robot are not systematically superior to those brought about by a human. In other studies, imitation training with a robot did not produce a significantly higher effect than training with a human [15,28]. Interaction with a robot may even induce a decrease in word and gesture imitation compared with interaction with a human [31,55]. The results are therefore mixed concerning a robot’s advantage in promoting imitation. It is important to note, however, that the studies that failed to observe an improvement in imitation with a robot were based on a single interaction session (with the exception of [31]). This suggests that a single session would not be effective in promoting imitation in individuals with ASD. Moreover, progress would vary according to the behavior to be imitated: while imitation of facial expressions is improved following repeated interaction with a robot, this is not the case for imitation of words or gestures [31].

The application of social robots with autistic individuals can also target the improvement of a particular behavior.

### 2.2. Fostering Specific Behaviors

The work presented in this section is of two types. Some studies promote the emergence of relevant and appropriate behaviors, while others aim to reduce maladaptive behaviors (stereotyped or repetitive behaviors) and anxiety. We will examine the benefits of a robot on these behaviors (see Table 2 for a summary).

#### 2.2.1. Supporting the Learning of Appropriate Behaviors

Interactions with robots offer the possibility of designing specific remediation to support particular learning, such as learning turn-taking [30,64] or non-verbal language gestures [20,65], emotion recognition [21], or regulating touch and physical contact [27].
behavsci-14-00131-t002_Table 2Table 2Benefits of robots in fostering specific behaviors in individuals with ASD.SectionArticleVariableEffectEffect *p*-ValueRobotSample Size (ASD)Mean Age (Standard Deviation)Functioning/Mean IQ (Standard Deviation)Duration of Robot Intervention (mn)CountryAppropriate behaviorsBharatharaj et al. [66]Touching interactionNANAKiliRo249.71 (3.24)NA1/day for 7 weeks (60 mn/session)India
Costa et al. [27]Appropriate touchgentle touch > harshp<0.05KASPAR86–10 y.o.NA7 sessions (NA)UK
David et al. [30]Turn-taking skillsrobot = human for 3 childrenp>0.05NAO53–5 y.o.LF-HF8–12 sessions (10 mn/session, 1/day)Romania
Ghiglino et al. [11]Theory of Mind skillstraining with humanoid robot > non-anthropomorphic robot and traditional therapyp<0.001; p<0.01iCub, COZMO435.8 (1.14)71.48 (16.50) (COZMO); 71.14 (15.49) (iCub)2/week for 8 weeks (15 mn/session)Italy
Holeva et al. [67]Theory of Mind skills (NEPSY II)robot training = human; pretest < post-testp>0.05; p<0.001NAO449.48 (1.95)IQ > 702/week for 3 months (NA)Greece
Lakatos et al. [68]Visual Perspective Taking and Theory of Mind skills (Charlie test)pretest < post-testp<0.05KASPAR138.11 (1.96)79.30 (14.33); range: 60–1031 to 10 sessions (15–20 mn/session)UK
Lee et al. [69]Proper force of touchingfeedback > no feedbackp<0.05Touch pad122 y.o.491 session (NA)Japan
Marino et al. [21]Recognition and understanding of emotionspretest < post-test (robot training); pretest = post-test (human training)p=0.001; p>0.05NAO1473.3 months (16.1) (robot group); 82.1 (12.4) (human)NA10 sessions (90 mn/session, 2/week)Italy
So et al. [65]Recognition and production of intransitive gesturesrobot training > no trainingp<0.05NAO305.10 (0.83) (experimental group); 5.8 (0.35) (control)NA4 sessions (30 mn/session, 2/week)China
So et al. [20]Recognition and production of emotional gesturesrobot training > no trainingp<0.001NAO138.99 (2.14) (experimental group); 9.50 (2.42) (control)range: 49–674 sessions (30 mn/session, 2/week)China
So et al. [70]Recognition and production of intransitive gesturesrobot training = humanp>0.05NAO239.17 (1.29) (robot group); 8.92 (0.93) (human)range: 46–745 sessions (30 mn/session, 2/week)China
Takata et al. [71]Understanding of others’ feelings and behaviorspretest < post-testp<0.01Sota, CommU, A-Lab android ST1417.57 (3.39)89.50 (10.95)5 sessions (1 h/session, 1/day)Japan
Wood et al. [72]Theory of Mind skills (Charlie test)pretest < post-test for 7/12 childrenp<0.05KASPAR1211–14 y.o.MA: 6–14 y.o.2–10 sessions (NA)UKReducing maladaptive behaviorsBharatharaj et al. [66]Stress levelpretest > post-testp<0.05KiliRo249.71 (3.24)NA1/day for 7 weeks (60 mn/session)India
Costa et al. [28]Stereotyped behaviorsrobot < humanp<0.05QTRobot159.73 (3.38)IQ < 80 (*n* = 8); 80–120 (*n* = 6); IQ > 120 (*n* = 1)1 session (1.5–4.3 mn)Luxembourg
Kumazaki et al. [50]Stress levelrobot < humanp<0.01ACTROID-F2929.1 (2.6)IQ ≥ 701 session (25 mn)Japan
Pop et al. [45]Stereotyped behaviorsrobot < humanp<0.05PROBO114–7 y.o.IQ > 708 sessions (1 mn/session)Romania
Shamsuddin et al. [32]Stereotyped behaviorsrobot < humanNANAO110 y.o.IQ = 1071 session (15 mn)Malaysia
Shamsuddin et al. [22]Stereotyped behaviorsrobot < humanNANAO68.9 (NA)range: 46–785 sessions (15 mn/session)Malaysia
Stanton et al. [48]Stereotyped behaviorsrobot < toyp=0.06AIBO115–8 y.o.NA1 session (30 mn)USAHF: High-Functioning Autism; IQ: Intellectual Quotient; LF: Low-Functioning Autism; MA: Mental Age; NA: Not Available; y.o.: Years Old.


Learning to take turns is necessary for playing with a partner, for example in a card game. However, children with ASD can have serious difficulties in grasping this behavior. Turn-taking training with a robot seems to be as effective as training with a human, although the children seem more interested in the robot [30]. It is all the more relevant when the child with ASD does not wish to cooperate with a human therapist [64].

Training with a robot can go hand in hand with learning to imitate intransitive or symbolic gestures. In the field of non-verbal communication, one observation is clear: children with ASD of both preschool (under 6) and school age (6–12) make fewer spontaneous gestures than TD children [73]. These gestures are fundamental to communication and interaction. It is therefore essential to understand and recognize them, and to be able to reproduce them in relevant contexts. A program using the NAO robot was developed for 30 children with ASD aged 4 to 6 with intellectual disabilities (IQ < 75) [65]. The program took place in two phases. In phase I, the participants learned to understand and recognize some twenty intransitive gestures (e.g., head nods and hand salutes), then learned by imitation to reproduce them (phase II). The performance of the group that received intervention with the robot was compared with that of a group that did not. The results show that the number of gestures recognized and imitated increased for the group of children who received training with the robot compared with the group who did not. This result was also found with training in the imitation of gestures expressing emotions [20]. Furthermore, the autistic children who trained with the robot even outperformed neurotypical children (i.e., TD children) of the same age on the immediate post-test, and performed comparably well on the delayed post-test [65]. They were also able to generalize these gestural skills to a new context, and this learning was stable two weeks after the post-test. It seems that gesture training with a robot is just as effective as with a human [70].

Individuals with ASD frequently present difficulties in attributing mental states (beliefs and emotions) to others, a capacity known as Theory of Mind (ToM) [74]. These difficulties, which persist throughout life [75], are observed both at the level of cognitive ToM, with failure on false belief tests [76], and at the level of affective ToM, with poorer abilities in interpreting facial emotions [77]. In children with ASD, the level of ToM predicts reciprocal social behaviors and the child’s subsequent social functioning: the more developed the ToM, the fewer social difficulties the child will have [78]. This ability is therefore an important factor in the development of children with ASD. Firstly, interaction with a robot in social and emotional situations would improve the child’s ability to understand the feelings and behaviors of others [71]. Secondly, this type of interaction could promote consideration of others’ mental states (i.e., ToM) in people with ASD. Training in visual perspective-taking with a robot is reported to enhance the performance of autistic children and adolescents on tests assessing ToM [11,67,68]. In a study of autistic children aged 5 to 11, visual perspective-taking training with a KASPAR robot resulted in better performance on the Charlie test, designed to assess ToM (but not on the Sally and Ann test, nor the Smarties test, also supposed to assess ToM) [68]. The authors point out that the latter tests are more difficult because they require complex mentalizing skills, unlike the Charlie test, which just requires inferring the direction of the character’s gaze. Nevertheless, this improvement following training with a robot is not necessarily observed in all children: in another study, only seven out of twelve children showed an improvement in their performance on the Charlie test (but not on the other tests). Moreover, statistical analysis revealed a significant difference when the results were analyzed as a group, but not at the individual level [72]. When intervention sessions aimed at training the social skills of children with ASD involved the assistance of an NAO robot, they generated improvements similar to those achieved by training with the therapist alone on the NEPSY-II social perception scale, corresponding to the ToM and emotion recognition subtests [67]. However, in two other studies, the presence of a robot increased the benefits of ToM training. Training with an iCub humanoid robot resulted in greater progress on the NEPSY-II ToM subscale than the same training with a human alone, or with a non-humanoid COZMO robot [11] (note that there was no difference in efficacy between the human and the non-humanoid robot). In cognitive–behavioral therapy with autistic children aged 4 to 8, with an NAO robot acting as co-therapist (its role is to provide emotional reinforcement and advice and tips), the results show a significant improvement in the recognition and understanding of emotions at the end of the intervention with the robot in contrast to the control group who trained with a human [21]. Progress was made not only in recognizing five basic emotions (anger, disgust, fear, happiness, and sadness) but also in more complex ones, such as shame. This improvement was also observed in the children’s homes, suggesting a generalization of the acquired behaviors.

Finally, people with ASD may react atypically in response to tactile sensations [79] and adapt their interpersonal touch less according to the situation compared with neurotypical people [69]. One study used a robot to develop sensitivity to physical contact(the robot’s sensory sensors are connected to a LED that lights up to encourage children’s physical interactions with the robot), but they did not analyze the results quantitatively [66]. Training with a robot would enable touch strength to be regulated, increasingly appropriate as sessions progress [27]. Feedback provided to the participant by a KASPAR robot (which presents schematic facial expressions) improves touch quality compared with no feedback [69].

In addition to using robots to increase the occurrence of certain behaviors, other protocols use robots to reduce the occurrence of maladaptive behaviors.

#### 2.2.2. Reducing Maladaptive Behaviors: Repetitive Behaviors and Anxiety

The use of the NAO robot could reduce the percentage of stereotyped behaviors in children aged 5 to 13 [22,32]. In other studies, children also show fewer stereotyped behaviors when performing an activity with a robot partner than with a human partner [28,45]. The frequency of autistic behavior tended to decrease when children interacted with an AIBO dog robot rather than a mechanical dog toy [48]. In addition, using a KILIRO parrot robot for 60 min a week for 7 weeks helps both children and adolescents (aged 6–16) to reduce their stress levels [66]. In adults with ASD, the stress reduction observed during job interview training was greater for training with an android robot than with a human [50], suggesting that interacting with a robot is less stressful than interacting with a human for individuals with ASD. It is worth noting that the most anxious children display more stereotyped behaviors [80]. Stress levels could be linked to the occurrence of global and motor stereotyped behaviors (but not to verbal stereotyped behaviors) [81]. Reducing the stress involved in interacting with a human by using a robot as a partner could therefore help to reduce stereotyped behaviors in individuals with ASD.

In this way, socially assistive robots can promote communication and social interaction in autistic children, support specific social learning, and reduce repetitive behaviors. However, the results are mixed when we compare the effectiveness of training with a robot with that of training with a human. While some studies show that robots are more effective than humans (e.g., [11,14,15,24,30,56]), others show that they are comparable (e.g., [13,28,67]) or even less effective [36,37] than humans. This heterogeneity in results (which will be discussed in more detail later; see Section 5.3) can be partly explained by the wide variety of skills targeted by robot intervention. A meta-analysis confirms the benefits of robots on the social development of children with ASD, but not on motor or emotional aspects [17].

## 3. A Preference for Interacting with Robots Rather than Humans in Individuals with ASD?

The benefits for individuals with ASD of interacting with a robot (at least in terms of social development) raise several questions. First, it is worth asking whether this type of interaction is more attractive to individuals with ASD than interaction with a human, as has been proposed in several studies [11,15,82,83]. In the next section, we will look at how individuals with ASD perceive robots.

### 3.1. Robots Could Be More Attractive than Humans to People with ASD

In general, it seems that individuals with ASD engage more in the task when faced with a robot rather than a human [60,84,85]. Some authors even argue that individuals with ASD prefer robots to non-robotic toys or humans [5,15,86].

Many studies have focused specifically on testing the preference of individuals with ASD for interacting with robots. People with autism do not show a human preference bias, unlike TD individuals who have more affinity with another human being rather than an artificial object [87]. The inversion effect of human faces(the phenomenon whereby human bodies—and faces—are recognized more quickly and accurately when presented in their usual orientation rather than upside down [88,89]) is less noticeable in individuals with ASD than in TD individuals [90] but similar for a cartoon face [91]. In autistic individuals, there is activation of the fusiform gyrus(the area involved in processing human faces in TD individuals) for robot faces but under-activation when processing human faces, unlike neurotypical individuals, whose activation is similar for all stimuli [92]. Furthermore, unlike TD adults, adults with ASD do not exhibit a gaze cueing effect with a human face(the gaze cueing effect refers to the tendency of TD individuals to direct their attention more quickly to an object when it is looked at by a person); i.e., they are no faster at detecting a target when it is looked at by a human versus not looked at. On the other hand, when the target is looked at by an EDDIE robot, they do indeed respond faster [43]. In TD individuals, conversely, the gaze cueing effect is less when a robot face rather than a human face is involved [93]. This suggests that TD individuals are more likely to follow human rather than robotic eye movements, while individuals with ASD follow robotic rather than human movements more [43]. Similarly, TD children prefer stimuli representing human biological movement, unlike children with ASD who show no preference [94,95,96]. At age 7–8, autistic children in the presence of a robot touch it more than neurotypical children [97]. Adolescents with autism report a higher level of pleasure than neurotypical adolescents when conversing with a robot (CommU and ACTROID F) [82]. They also tend to confide more with a robot (CommU) than with a human regarding embarrassing experiences [82]. Conversely, TD individuals’ tendency to confide is not improved when interacting with the robot. Finally, in adults, autistic individuals judge a synthetic voice and a human voice equally positively, unlike neurotypical individuals who prefer the human voice [98]. According to this study, this result is due to the fact that autistic individuals would be less sensitive to the subtle differences between the two voices.

Children with ASD show a more pronounced physiological response (increased heart rate) when interacting with a robot than with a human [99]. It therefore seems appropriate to test directly whether autistic people prefer to interact with a robot rather than a human. In other words, if young people with autism have to choose between a robot and a human for an interview, which interlocutor will their choice lean towards? To answer this question, 23 participants with ASD aged 17 to 25 with an IQ ≥ 70 were recruited [84]. In a single 10-minute session, half of the ASD participants were interviewed by a robot (ACTROID-F) followed by a human, while the other half were interviewed in reverse order. The robot was teleoperated. The interview focused on assessing the robot’s degree of humanity on different characteristics: humanness (1 = mechanistic, 5 = human-like), emotional (1 = lacking in emotion, 5 = rich in emotion), animatedness (1 = inanimate, 5 = animate), naturalness (1 = unnatural, 5 = natural), familiarity (1 = unfamiliar, 5 = familiar), warmth (1 = cold, 5 = warm), complexity (1 = simple, 5 = complex), and regularity (1 = random, 5 = regular). This was followed by three questions: “Would you like to practice interviewing with this robot again?”, “Do you think interviewing with this robot makes you competent?”, and “How motivated are you to train in interviewing with this robot compared to a human?”. Firstly, the robot was well accepted: in response to the first two questions, the participants answered “Yes” to 87%. Regarding the third question, the participants showed greater motivation to undergo interview training with the robot than with a human. Moreover, this motivation is negatively correlated with the robot’s impression of humanity, i.e., its perceived resemblance to a human. In other words, people with ASD who reported a reduced impression of the robot’s humanity were the most motivated to undergo interview training with a robot rather than a human.

Robots therefore appear to be attractive stimuli for autistic individuals, more so than humans or inanimate objects, and provide benefits to this population. In contrast, this result is not observed in TD individuals. Nevertheless, it may be an exaggeration to speak of a preference for robotsamong people with ASD.We have previously used the term “preference” following some studies [5,15], but the term “increased interest” is more appropriate since people with ASD seem to direct their attention more towards robots than towards humans, without it being possible at present to determine whether this is the result of explicit decisionmaking (in which case we could speak of preference [100]). From now on, therefore, we will use the term “increased interest”.

Several explanations have been put forward to explain the interest of people with ASD in robots. A first avenue that could be explored is that of a different categorization of robots, or a different tendency towards anthropomorphism in people with autism.

### 3.2. Difference in Robot Categorization and Anthropomorphism between Typical Development and ASD

It is possible that people with ASD view robots differently from people with TD. We will now focus on the differences in the way robots are perceived and the different attributions made to them.

#### 3.2.1. Difference in Robot Categorization between Typical Development and ASD

The majority of TD children perceive the robot as a social being deserving of fair treatment but do not believe it possesses its own freedom or civil rights [101]. TD children can therefore attribute certain human characteristics to a robot without considering it completely as a human. The robot is not perceived as an animate or inanimate entity but as belonging to a new ontological category, possessing particular properties [102]. However, autistic children’s perception of robots may differ from that of TD children. Indeed, the processing of robot features (both appearance and behavior) depends on the user’s cognitive abilities [103]. Individuals with ASD have cognitive impairments, which means they will not necessarily interpret cues in the same way [60].

The way children with ASD categorize robots seems comparable to that of TD children. ASD and TD children aged 5 to 7 mostly perceived robots (NAO, ROMIBO, KEEPON, PROBO, PLEO, and KASPAR) as toys (although some autistic children also associated them with a machine, particularly boys) [104]. Similarly, in another study, the NAO robot is perceived as a toy by autistic and TD children aged 5 to 7, with no significant difference in robot categorization between groups [105]. Nevertheless, these categorization studies were not carried out using the same methodology: one involved categorizing the images of several robots after seeing them in a video [104], while the other involved categorizing a specific robot (NAO) after interacting directly with it [105]. Yet, the way in which the robot is presented (in a photo, a video, or during a virtual or physical interaction) can produce different effects on its perception, as has been highlighted in the TD population [106] but also in the ASD population [107]. The second categorization study therefore follows a more ecological methodology, although it focuses on a single robot. Moreover, it is important to note that the perception of robots is variable among autistic individuals, since some children interact with robots as objects, while others treat them as social agents [108]. This variability could be due—at least in part—to the fact that the robot is both an object and a social agent, “treated like a living thing while it is handled as a material thing” [109] (Alač 2016, p. 553). We could define a social agent as an entity capable of emitting and receiving social behaviors, and, even if the robot does not act autonomously, it provides the illusion of doing so. As a result, although people are aware of the limits of the robot’s autonomy, their engagement in interaction with a robot is analogous to that observed during human-to-human interactions, and they treat the robot like a participant in its own right [110] (even going so far as to apply human social norms to it [111,112]). In this context, a social robot can be considered as a social agent with which individuals interact, although this type of interaction is not as rich as interaction with a human. The interindividual variability inherent in ASD will also be discussed later (see Section 5.2), but we can note that robots are generally well accepted by autistic children [113]. Regarding the categorization of robots by people with ASD, further studies are needed.

Although autistic children seem to categorize robots in a similar way to neurotypical children, they are nevertheless likely to attribute different characteristics to them.

#### 3.2.2. Difference in Robot Anthropomorphism between Typical Development and ASD

TD adults are prone to anthropomorphism; i.e., they tend to attribute human characteristics to non-human entities, such as robots [114,115]. TD children are also affected by this phenomenon as they attribute desires and physiological states to robots [116], interpret their movements as goal-directed [117], and seek to help them [118]. Insofar as autistic and neurotypical children categorize robots in similar ways, we might wonder whether they also exhibit the same level of anthropomorphism.

We mentioned the ToM deficit present in many autistic individuals [74]. The concepts of ToM—the attribution of mental states to a human—and anthropomorphism—the attribution of human characteristics to a non-human agent—are thought to be closely related [119,120]. TD individuals would use their ToM expertise to predict the behaviors of non-human agents in a bid to explain their environment effectively. Given the ToM deficit present in autism, studies suggest that anthropomorphism may be reduced in individuals with ASD [114,119]. The difficulties encountered by individuals with ASD in attributing mental states to others would also apply to non-human agents, resulting in fewer attributions towards these agents than in neurotypical individuals.

However, according to other studies, anthropomorphism cannot be considered an extension of ToM [121], and individuals with ASD may instead show an increased tendency towards anthropomorphism. In the general population, the number of autistic traits individuals possess is positively correlated with anthropomorphism: the more autistic traits participants possess, the more likely they are to attribute human characteristics to non-human agents [122,123,124]. A study of adults with ASD also indicates an increased tendency to anthropomorphize objects compared to TD adults [125]. Adults with ASD report a similar impression of humanity for a synthetic voice and a human voice in contrast to neurotypical individuals [98]. Autistic individuals also seem less sensitive to the physical irregularities of non-human agents: adolescents perceive an android robot (Actroid-F) as more human-like than TD adolescents [126]. Children with ASD and TD children attribute fewer human characteristics to a robot than to a human, indicating that they differentiate between them. However, the difference in perceived humanity score between robot and human tends to be smaller in children with ASD than in TD children [127]. Children with ASD are more likely to think that a robot can grow up than TD children and are more likely to think that a robot can feel pain, although this latter tendency remains marginal [128]. Moreover, they are less distrustful of the robot than TD children when it provides incorrect information, and even less so when the degree of anthropomorphism is high(in TD children, no correlation is observed between the degree of anthropomorphism and distrust of the robot). This indicates that the perception of the robot as human-like impacts autistic children’s interactions with the robot and through mechanisms potentially different from those involved in typical development.

It would be possible to explain this increased tendency towards anthropomorphism by drawing on the theory of [114], which states that this phenomenon is modulated by various psychological determinants, including the motivation to explain and understand the behavior of other agents (effectance motivation) and the desire for social contact and affiliation with others (social motivation) [120]. Yet, we shall see that the restricted interests and social difficulties encountered by individuals with autism influence these motivations.

##### Restricted Interests: Effectance Motivation

People with ASD show significant deficits in the recognition and processing of human stimuli (faces, gaze, and biological movements), and less neurohormonal reward in social interaction with others compared with neurotypical people [129]. As early as 2 years of age, TD children show a preference for human rather than non-human movement, but autistic children do not [95]. On the contrary, these children express greater interest in non-human stimuli than neurotypical individuals, which may be linked to the restricted interests manifested in ASD. Indeed, 75% of people with autism show a major interest in a very restricted domain [130]. Among the most common, we can cite interest in luminous or rotating objects (e.g., focusing on the rotation of a washing machine drum), in cartoon characters, in animals, in computers, or in mechanical objects [130]. Yet, the characteristics of non-human agents can refer to the individual’s restricted interests (e.g., cartoons [131], animals, and mechanical movements [132], or robots and computers [18]). Given that anthropomorphism is linked to the desire to understand an agent [114], autistic children’s interest in non-human agents or objects could increase their tendency towards anthropomorphism. They would rely on anthropomorphism to facilitate understanding and the prediction of future behaviors of this attractive agent. Anthropomorphism would then serve to reduce the uncertainty of the environment, resulting in minimized anxiety [125].

##### Social Isolation: Social Motivation

Contrary to what one might think given their lesser involvement in social interactions, people with ASD need to create social links with others [133]. However, they are particularly vulnerable to social isolation, which is thought to be caused by their difficulties in social understanding rather than a desire for solitude [134]. In the general population, individuals with the most autistic traits also report the highest rate of isolation [134]. People with ASD report more feelings of loneliness and loneliness-related stress than TD individuals [135,136]. Yet, social isolation would increase the tendency to anthropomorphize objects [114], such as alarm clocks [137], telephones [138], and robots [139,140]. Anthropomorphism would thus reduce the feeling of social disconnection [125], which is why it would vary depending on the individual. Conversely, when individuals are encouraged to recall intimate social relationships, their tendency to anthropomorphize decreases [137]. Anthropomorphism would therefore play the role of a compensatory mechanism for people with a lack of social connection, as is the case with people with ASD [123]. Indeed, among adults with ASD, those who report high levels of loneliness attribute more human characteristics to objects [123].

Thus, people with ASD would anthropomorphize more than TD individuals to reduce uncertainty in their environment and manage the loneliness felt in everyday life [123,125]. Interactions with robots would then be more socially motivating for autistic individuals: not only do anthropomorphic stimuli increase theireffectance motivation but they could also meet their need for social connection [120].

#### 3.2.3. Improving ToM of Individuals with ASD with a Robot

Individuals with autism would even show better ToM skills when confronted with cartoon-like or animal-shaped characters rather than human ones [120]. This is also found with robots [54]. Several studies seem to confirm these observations.

Neurotypical adolescents performed better than autistic adolescents on an emotion recognition task involving a human face. Nevertheless, when the face was that of a cartoon, the two groups showed comparable performance, implying an improvement in the performance of the autistic adolescents(although their accuracy in the cartoon face condition remained lower than that of the neurotypical group in the human face condition) [141]. This improvement was also observed when emotions were presented on an animal rather than a human face: adolescents with ASD were then able to recognize them better [142]. Similar results were obtained with adults selected from the general population and then divided into two distinct groups according to their autistic quotient: a group with few autistic traits, and a group with a high number of autistic traits [143]. (The autistic quotient is a tool designed to assess the number of autistic traits in an individual, i.e., characteristics reminiscent of the functioning of an individual with ASD. It does not constitute a diagnostic scale but enables the individual to be situated on the continuum between ASD and normality [144]). Participants then took the Reading the Mind in the Eyes Test (RMET), which involves identifying the emotions expressed in different photos of faces, a task in which people with ASD generally do not perform as well as neurotypical people [77]. This study involved both the original version of the RMET (with photos of faces) and a modified version (with drawings of faces). The results showed that, in the original version, the group of participants with a high number of autistic traits failed more than the other group, while, in the modified version, no difference was observed. Finally, in another ToM task (social faux pas), in which the performance is negatively correlated with the number of autistic traits, we note that this correlation disappeared in an adapted version with animals [145]. In other words, in the adapted version, no difference in performance is noted between participants with a high or low number of autistic traits. Moreover, when autistic children aged 4 to 7 undergo ToM training with non-human agents presented in the form of a film(children watch the animated series “The Transporters” designed to support emotion recognition and understanding, which features different vehicles with humanoid faces), their performance increases [146,147]. This improvement also applies to novel anthropomorphic stimuli and generalizes to human stimuli [146] up to 3 months after the end of the intervention [147].

A study comparing facial emotion recognition training with a robot and with a human shows a significant improvement in emotion recognition and attribution between pre-test and post-test with the robot but not with the human [54]. At the end of the intervention, the children who trained with the robot were better at identifying the emotions felt by the characters in a story than the children who trained with the human. These results suggest that, in individuals with ASD, emotion recognition is facilitated when it involves a non-human agent. Similarly, interaction with a robot around social and emotional situations generates an improvement in consideration of others’ feelings and behaviors [71]. We observed that visual perspective-taking training with a KASPAR robot improved the performance of children with ASD on the Charlie test, designed to assess ToM (but not on the Sally and Ann test, nor on the Smarties test) [68]. ToM training with a humanoid robot would also result in more progress on the NEPSY-II ToM subscale than the same training with a human or a non-humanoid robot [11]. Nevertheless, no difference in efficacy was found in this study between the human and the non-humanoid robot, nor in another study conducted with the NAO robot [67]. Therefore, the use of non-human stimuli could promote social–cognitive development and support the transfer of these skills to human stimuli [120], and potentially more effectively than with a human stimulus [11]. However, it is important to note that the benefits provided by robots can vary according to their appearance (this point will be discussed in Section 5.1).

## 4. Reasons Provided to Explain the Benefits of Robots

In the literature, we noted potentially increased anthropomorphism in people with ASD, as well as better ToM skills with robotic stimuli—and better social skills overall [17]. Training with a robot seems to reduce social difficulties more than training with a human [11,54], but why do autistic people seem to mobilize their social skills better in the presence of a robot? Several explanations can be proposed based on different theories aimed at explaining the social difficulties observed in autism. We have already observed that people with ASD have social difficulties and show less interest in social stimuli than TD individuals. The link between social skills and social motivation seems well established in children with ASD: those with the least social motivation have more severe social difficulties [148,149,150]. Nevertheless, the meaning of this association remains to be determined.

Two main theories have been proposed to explain the association between social difficulties and reduced social interest observed in autism [151]: Social Motivation Theory [152] and Social Cognition Theory [74,153]. We will analyze the two theories that seek to explain the causality between these two characteristics of ASD and build on these theories to explain the benefits brought about by robots.

### 4.1. Social Motivation Theory and Robot as Motivator

#### 4.1.1. Social Motivation Theory

Social Motivation Theory assumes that the initial deficit of people with ASD is in social motivation [152]. Social motivation refers to the set of psychological dispositions and biological mechanisms that induce individuals (1) to orient themselves preferentially towards the social world (a component called social orienting); (2) to seek out social interactions, and take pleasure in them (a component called social reward); and (3) to foster and maintain social ties (a component called social maintaining). Individuals with ASD show deficits in each of these components.

Firstly, people with ASD orient themselves less towards social information [152,154]. They look less at people, and, during visual exploration of a social image, they pay less attention to social stimuli (eyes and mouth) than TD children [57,58] and more to non-social stimuli [155]. These studies suggest less engagement in the processing of such information and thus in social learning. Secondly, individuals with ASD seem less sensitive to social rewards than neurotypical individuals [156,157]. For example, they score lower on friendship questionnaires designed to assess the pleasure experienced in this type of relationship [158] and report feeling less pleasure in social situations [159]. This degree of social anhedonia (i.e., loss of the ability to experience social pleasure) is positively correlated with the degree of ASD severity [159]. Thirdly, people with ASD show a deficit in social maintaining: for example, they are less likely than neurotypical people to use greeting or farewell gestures [160], or to resort to social maintaining strategies such as affect concealment [161] or social laughter [162].

In typical development, social motivation is crucial for the development of social skills. However, in children with ASD, altered social motivation would impact social abilities and behaviors through cascading effects [149]. Indeed, the reduced interest in social environments would prevent children with ASD from being attentive to socially relevant stimuli in their environment and thus from accumulating rich social experiences, which would consequently provoke difficulties in social cognition.

Several studies seem to agree with Social Motivation Theory (see [157] for a review). The tendency of children with ASD aged between 1 and 5 years to orient themselves preferentially towards a social image (rather than a geometric image) is positively correlated with the frequency of joint attention behaviors [57]. What is more, the degree of social motivation in one-year-old children with ASD predicts functional language development at age 3: the more socially motivated a child is at age 1, the more intentional acts of communication he or she will produce two years later [163]. Yet, robots could have an impact on the social motivation of children with ASD.

#### 4.1.2. Robot as a Motivator

We have observed that individuals with ASD generally show better social skills when interacting with a robot. The first hypothesis is that robots are more motivating than humans for people with ASD. If they are indeed more motivated to interact with robots, this type of interaction would enable them to train their social skills more effectively than in the context of interaction with a human, and consequently reduce their social difficulties.

Children with ASD report positive affect after interaction with a robot [113] and even an improvement in their emotional state after interaction with a PELOPPA robot [99], which seems to confirm that interaction with a robot is pleasant for these children. When comparing interaction with a robot to interaction with a human in children with ASD aged 4–12 years, they report more positive affect with the robot [164]. The sensory reward provided by the robot would evoke more positive reactions than verbal encouragement provided by humans [165], which could be related to the specific interests of autistic children who often enjoy technology and robotics [24]. The robot could then be used as a reinforcer [44,164], as a social mediator [166,167]—in other words, “a tool that mediates (encourages and facilitates) social behaviors among children, and among children and adults ” [166] (Werry 2001, p. 58)—or as a social motivator [24]. If social interaction with a robot is synonymous with reward for individuals with ASD, this could promote the association between social interactions and reward [44]. By increasing the motivation to interact in individuals with ASD, thanks to robots, it would be possible for them to access more social experiences (even if these are simplified compared to social experiences with humans), which would have the consequence of enriching their understanding of the social world and therefore reducing the difficulties encountered in this area.

However, another theory could also explain the pattern of results obtained when autistic individuals interact with a robot.

### 4.2. Social Cognition Theory and Robot as Simplified and Predictable Social Agent

#### 4.2.1. Social Cognition Theory

According to Social Cognition Theory (“mindblindness” account), people with ASD present a deficit in their ability to understand the social world, which then leads to a reduction in interest in social situations [74]. More precisely, this initial deficit would concern social cognition, notably ToM, and would then impact social motivation. People with ASD would turn away from the social world as a result of the difficulties they face in understanding it. In this theoretical framework, reduced social motivation can be conceptualized as a strategy for compensating for social difficulties: individuals with ASD would avoid social situations to protect themselves against the mental suffering associated with social rejection [168]. Indeed, in conversations, adults with ASD are evaluated less positively by their partners than TD adults [169], and neurotypical observers express reduced intentions to interact with them [170].

A more advanced theory by the same author has also been proposed to explain the non-social symptoms (repetitive behaviors and restricted interests) of autism: the Empathizing–Systemizing Theory [153]. It takes up the previous theory and postulates that, while individuals with ASD show a reduced tendency to attribute mental states to others (empathizing) compared to the average, which would explain their social symptoms (deficits in communication and social interaction), on the contrary, they show better-than-average abilities to spot the regularities underlying systems and analyze these rules (systemizing). This latter aspect would explain the restricted interests and repetitive behaviors of individuals with ASD, as well as their aversion to change: their preference is for predictable systems. Restricted and repetitive behaviors would be the consequence of an inappropriate coping strategy aimed at reducing anxiety or negative affect by trying to control the environment [80]. The non-social symptoms of autism would then result from the desire of individuals with ASD to maintain a predictable environment. Indeed, people with autism frequently display an intolerance of uncertainty, and this characteristic could moderate the link between autism and anxiety [171].

Within this theoretical framework, we will see that robots could impact social difficulties in two ways.

#### 4.2.2. Robot as Simplified Social Agent

The second hypothesis put forward in the literature to explain the benefits of robots is that they make social interactions easier to understand. Communicating with a robot would be less complex than communicating with a human [85,172]: robots would be simpler social agents than humans, making social contact easier [173]. The mode of social interaction they enable would be simpler to understand [30,60] and could be less intimidating for individuals with ASD [174]. Robots would represent a stage of intermediate difficulty compared with humans (i.e., a stimulus less complex to understand than a human), and could thus facilitate interactions by making them more accessible to the child, even if they involve other people. This would explain why a child’s interaction with an adult is simpler in the presence of a robot despite the apparent increase in the amount of information to be processed in the interaction situation—which involves three agents instead of two. Child–robot interaction would remain simpler than child–robot–adult interaction both quantitatively because it involves fewer agents and qualitatively because the agent involved is simpler to understand). As simplified social agents, robots could reduce the social difficulties of people with ASD, thereby increasing interest in this type of interaction. This reasoning seems congruent with the results obtained.

Firstly, a robot’s perceptual processing is less complex than a human’s as it has fewer social cues [175]. Yet, autistic children can feel overwhelmed by the information contained in a human face [176]. For individuals who have difficulty understanding emotions, gestures, or facial expressions, the simplified social behavior of robots would be particularly beneficial [4], not least because they have a simpler body and face than a human [177]. The simplicity of the robot would enable children with ASD to better mobilize their skills in a social learning context: they learn more quickly to distrust an agent who provides incorrect information when that agent is a robot rather than a human [128]. This would explain why individuals with ASD show better ToM skills when training with a robot rather than a human [14,21,54]. Interaction with a human can even cause confusion or stress in autistic children [178]. However, interaction with a robot (in the presence of a human) would lead to less social anxiety than interaction with a human alone [50] and could reduce the stress felt [167]. This simplified interaction would then allow more self-disclosure, particularly on negative topics, and adults with ASD report less embarrassment during the interaction [179]. Given the link between stress and stereotyped behavior [80,81], Social Cognition Theory could explain why interaction with a robot minimizes the occurrence of stereotyped behavior if this result is confirmed by the scientific literature.

#### 4.2.3. Robot as a Predictable Agent

Moreover, the predictable nature of the robot’s behavior also means that this interaction is easier to understand than an interaction with a human [30,180,181]. With Empathizing–Systemizing Theory, we saw that the decreased motivation towards social stimuli in autistic people could be explained by an aversion to unpredictability. A stimulus that is easier to understand because it can be predicted would then be more motivating. Social interactions with robots are more predictable than interactions with humans [11,177,180] because robots are more predictable systems than humans. Robots could thus play the role of an intermediate stage in the development of social interaction [182]. Furthermore, the predictable nature of robot behavior may also reduce the anxiety experienced during interaction [50], which could explain the reduction in restricted and stereotyped behaviors observed when interacting with a robot compared to interacting with a human [28,45]. Indeed, a high level of anxiety would be associated with the increased occurrence of restricted and stereotyped behaviors [80,81].

Experimental results seem to be in line with this theory. A study varying the predictability of the robot’s behavior over time showed that the visual attention of children with ASD was less fixed on the activity when the robot behaved unpredictably (i.e., it is programmed never to perform the same behavior in two sessions in a row, with variations in words spoken, intonation, movements, etc.) rather than predictably [180]. No differences were found on other measures of behavioral engagement. This reduction in the attention paid to the robot when it behaves unpredictably can be viewed as a coping strategy aimed at reducing the anxiety felt as a result of the inability to predict the robot’s actions [180]. Similarly, a robot with contingent behavior (which responds to the child’s behavior) evokes more social behavior than a robot with non-contingent behavior (which responds randomly) [12]. Nevertheless, a robot programmed to perform unpredictable behaviors would still be easier to predict than a human as the variation in its behaviors is smaller [180]. The predictability of the activity itself also seems to play a crucial role: children engage and watch the robot more when the interaction session involves familiar rather than unusual or novel activities [47].

### 4.3. Criticism of These Theories

Nevertheless, both theories are currently the subject of criticism. Indeed, the social motivation and social cognition abilities of adults with ASD are not predictive of the social behaviors exhibited in real-life interaction [183].

Concerning Social Motivation Theory, the results are mixed [151]. While some studies seem to confirm it (see [157] for a review), others present contradictory results. Firstly, individuals with ASD report a higher sense of loneliness than TD individuals [136], suggesting that they show a motivation to create social bonds. The compensatory strategies put in place by some individuals with ASD to mask their difficulties, i.e., to present neurotypical behaviors despite the social difficulties encountered, would also indicate preserved social motivation, at least for part of this population [168]. There would therefore be a lack of universality in the social motivation deficit [184], with interindividual variation in social motivation [150,151]. Moreover, social motivation does not predict attention to faces in adults with ASD [185]. Furthermore, the specificity of the social reward deficit is questioned: people with ASD would show an atypical reaction in response to social rewards, but also in response to non-social rewards [151,186]. Finally, Social Motivation Theory seems incomplete as it does not explain the non-social symptoms of autism (restricted interests and stereotyped behaviors) [168,184], unlike the Empathizing–Systemizing Theory [153].

Although it seems to explain both social and non-social symptoms, Social Cognition Theory (which maintains that social difficulties are due to an initial deficit in ToM) has also been criticized [187]. Firstly, this deficit would not be specific to the autistic population but would also concern individuals with other disorders. It would not be universal since there is some variation from one autistic individual to another in social cognition abilities, with some individuals passing ToM tests. Lastly, the studies highlighting this deficit are not necessarily replicated, and ToM tests do not have predictive or convergent validity.

Moreover, several studies suggest that deficits in social motivation and social cognition are not present at an early age. Thus, at the age of 6 months, the social behavior (looking at faces, social smiles, and vocalizations) of children subsequently diagnosed with ASD is comparable to that of TD children, whereas, at 12 months, children with ASD produce less social behavior than TD children [188], and they look less at faces at 18 months [189]. Furthermore, the degree of social motivation observed at 6 months was not associated with the initiation of joint attention at 12–15 months [190]. However, the greater the increase in social motivation between 6 and 12 months, the more the child will initiate joint attention behavior at 12–15 months. This developmental period would correspond to an increase in social engagement in TD children but not in children with ASD (who could even, on the contrary, show a reduction in social engagement) [190]. Similarly, a study looking at eye gaze duration shows that it decreases between 2 and 6 months in children with ASD, whereas it increases in TD children [59]. Thus, joint attention and eye contact would not be affected initially in ASD but would become so through cascading effects following an interaction between social motivation and emerging socio-cognitive abilities. It is possible that cerebral differences exist from the first year of life in children with ASD but that they only become visible when more complex treatments are required to express behaviors appropriate for the age group [191]. In addition, the age at which deficits emerge varies considerably from one child to another [192].

While social anhedonia (linked to social motivation) predicts the diagnosis of autism and the number of autistic traits, ToM (assessed here by the RMET) does not [193]. This result therefore seems more consistent with Social Motivation Theory than with Social Cognition Theory. However, it is possible that difficulties in social cognition and social motivation are interdependent. Insofar as they are underpinned by common neuronal substrates (at least partially), they can be conceptualized as forming part of the same developmental axis of the social symptoms of autism [194]. Social motivation skills would interact with social cognition skills [190]. However, they do not completely overlap since the degree of social motivation is not correlated with the performance obtained on a ToM test [96], nor with prosocial behaviors such as spontaneous help [195]. Consequently, it remains complex to identify the meaning of this link at present.

The robot may act both on social motivation by increasing children’s interest in activities involving the presence of a robot and on difficulties in understanding the social world since it is a simplified stimulus compared with humans, making its behavior easier to interpret. The predictability of the robot’s behavior could also make it easier to understand and more motivating for individuals with ASD.

Another hypothesis remains to be explored. The fact that people with ASD mobilize their abilities better in the presence of a robot could also be explained using the “double empathy problem” framework [196,197], a theory that postulates that people with ASD understand interactions better when interacting with people with ASD, while neurotypical people understand interactions better when interacting with neurotypical people. People with ASD would encounter difficulties in interacting with neurotypical people—and vice versa—due to a “disjuncture in reciprocity between two differently disposed social actors” [196] (Milton 2012, p. 884). Within this theoretical framework, it could be argued that robots, through their impoverished expressions, are closer to a type of interaction exhibited by people with ASD than a type of interaction exhibited by neurotypical people, which would then enable a better understanding of these interactions because the disjuncture in reciprocity evoked in [196] would be less. However, we do not believe that the expression of an impoverished robot is similar to the expression of a person with ASD. This theory should therefore be treated with great caution.

In short, interaction with robots is no substitute for interaction with humans. However, robots, as attractive, simplified, and predictable social agents, could help individuals with ASD to understand the logic underlying social interactions, and this understanding could then be generalized to situations encountered in everyday life [198]. Nevertheless, several challenges remain to be resolved in the field of interaction between autistic children and robots.

## 5. Current Challenges in the Interaction of Autistic Individuals with a Robot

Some studies question the engagement usually observed in the activity as some autistic children show a low or non-existent level of engagement in the presence of a robot [199,200,201], or even a loss of attention occurring rapidly during the activity [29,34]. Furthermore, the increase in learning performance with a robot is not systematic despite the child’s involvement in the task [33,34,200]. However, these differences in results can be explained, at least in part, by the variability in the type of robot used, the heterogeneous symptomatology of autism, and methodological limitations. These elements will be discussed in this section.

In TD individuals, the type of robot used plays a role in its perception and the behaviors expressed towards it, but the characteristics of the user and the context of the interaction (the situation in which it takes place) also determine how it unfolds [106]. It is therefore important to take them into account when analyzing the results. Similarly, in individuals with ASD, the appearance of the robot [167], the profile of the child [202], and the intervention context can also have an impact on the interaction.

### 5.1. Preference by Robot Type: From the Uncanny Valley to the Uncanny Cliff

Firstly, we might ask whether the benefits brought by robots to individuals with ASD vary according to the type of robot used.

#### 5.1.1. Different Robot Appearances

Indeed, it is important to note a great variety of appearances in the robots used with autistic individuals [203], and a lack of consensus as to which robot should be preferred [17,178]. Three categories of social robots were distinguished according to their type of design: abstract, iconic, and humanoid [204]. In this taxonomy, abstract robots refer to robots that have no human morphological elements and whose appearance is mechanical (e.g., LEGO MINDSTORM). Iconic robots generally have certain physical human characteristics—such as eyes, mouth, and arms—but their appearance is still highly mechanical (e.g., NAO). Humanoid robots have a very human-like appearance (e.g., SOPHIA). We can add two categories for a more exhaustive taxonomy. Android robots are robots that resemble humans in appearance and behavior (e.g., ACTROID-F), while zoomorphic robots imitate an animal form (e.g., PARO).

Numerous robots of varying appearance are frequently used in the field of autism: ROBOTA, INFANOID, KEEPON, KASPAR, NAO, CommU, ACTROID-F, and PLEO (ranked in ascending order of degree of freedom) [85]. Generally speaking, the NAO robot is the most widely used for interaction with autistic children [16,205] and is well accepted by this population [32]. Although the majority of studies use a robot resembling a human [205], zoomorphic robots such as PLEO the dinosaur, AIBO the dog, or PARO the seal constitute another interesting modality for meeting the needs of individuals with ASD. An intervention based on a game with a parrot robot (KILIRO) reduces the stress levels of both children and adolescents (aged 6–16) [66]. The AIBO robot also increases the engagement of children with ASD (aged 5–8) in interactions, including verbal behavior [48]. It therefore remains to be determined which appearance of the robot is optimal for interaction with autistic individuals.

For TD individuals, the robot’s appearance is of crucial importance for interaction: the more a robot resembles a human, the more mental abilities will be attributed to it [206]. Similarly, individuals with ASD may behave differently in interaction depending on the robot’s appearance. We have observed that autistic adolescents report a higher level of pleasure than neurotypical individuals when interacting with a robot, whether its appearance is moderately human-like (CommU) or strongly human-like (ACTROID-F) [82]. Nevertheless, while individuals with ASD showed an increased tendency to confide in the iconic robot CommU compared to interacting with a human, this benefit was not observed with the android robot ACTROID-F. Children with ASD seem to perform more social behaviors when interacting with a robot with mechanical characteristics than with a robot with human characteristics [86]. More specifically, this study involved two types of agents: ROBOTA, a robotic doll, and a mime that performed the same gestures as the robot. Each of these agents could be presented with their face uncovered and wearing human clothes (human appearance condition), or with their face under a mask and wearing a metallic gray suit (robotic/mechanical appearance condition). Children interact more with the agent when it has a more mechanical appearance regardless of whether it is a robot or a human.

However, other studies point to an appeal for implementing human features on robots. Firstly, parents of children with ASD report that toys with a face, or moving limbs, are more supportive of the development of social skills, including facial expressions [207]. An NAO robot presented in a humanized way with clothes leads to more interest and positive affect [208]. Furthermore, a humanoid iCub robot provides better ToM benefits than a non-human-like COZMO robot [11]. This suggests that robots with certain human characteristics could be more effective in supporting the social development of children with ASD.

On the one hand, as autistic children have difficulty transferring their social skills to contexts other than the one in which they learned them, it seems relevant to use the most human-like robots as humanoid and android robots have the advantage of promoting the generalization of acquired behaviors into everyday life [85,203]. Thus, the improvement in ToM performance is significantly greater with a humanoid robot than with a non-humanoid robot or a human [11], suggesting that the proximity between the interactions enabled by the robot and the social interactions of everyday life does play a role in the effectiveness of interventions. On the other hand, less human-like robots—iconic or zoomorphic—are well accepted as they facilitate engagement in interaction, and a simplified agent with exaggerated social cues can help children with ASD to focus on important social information. It is thus difficult to provide a clear answer to the question about preference according to robot type: the results are not clear-cut.

The preference of children with ASD has been compared with that of TD children. A study comparing several robots (KEEPON, NAO, KASPAR, PROBO, PLEO, and ROMIBO) first highlighted a similar preference for autistic and neurotypical children aged 5–7 towards the human-like robot KEEPON (the children were asked to rank the six robots, starting with their favorite and ending with their least favorite, and the KEEPON robot is the one most often chosen in the first three positions) [104]. Nevertheless, the robot that comes second for neurotypical children (PLEO—dinosaur robot) is among those that children with ASD like least. Conversely, the KASPAR robot, the most human-like, is one of the least appreciated by neurotypical children, while half of autistic children ranked it among their three favorites. So, when asked to choose between different robots, autistic and TD children do not necessarily show the same preferences. The uncanny valley phenomenon could explain this result.

#### 5.1.2. From the Uncanny Valley to the Uncanny Cliff

In TD individuals, a human-like robot is perceived more positively than a less human-like robot. However, beyond a certain degree of human resemblance, the robot triggers a feeling of unease or strangeness in the user, a phenomenon known as the uncanny valley [209,210,211]. This term refers to the aspect of the relationship curve between the level of sympathy felt by the user and the degree of human-likeness of the robot. Neurotypical individuals thus feel a sense of strangeness when faced with a robot with a strongly human appearance [212] and prefer a weakly to moderately human-like appearance [213]. An explanation has been proposed for the uncanny valley: the human appearance of robots would induce expectations in the user [214] (in terms of its capabilities, for example) and this resemblance would encourage individuals to apprehend the robot according to human normative expectations [215]. Yet, the divergence of the robot’s behavior from the human norm would make it appear frightening and would be at the origin of this phenomenon [209,210,211]. The uncanny valley is observed in TD people, but we wonder whether this phenomenon is also observable in autistic people given their preferences, which diverge from those of neurotypical people.

Based on a Bayesian model, a theoretical paper argues that robots would induce a unique emotional response curve in autistic children [216]. The positive emotional response curve—which takes the form of a valley in neurotypical children, with a trough for robots strongly resembling humans, then an increase for humans—would resemble a cliff more than a valley in autistic children; i.e., the response would not increase for humans. So, children and adults with ASD would show a reduced tendency to the uncanny valley [95,217]. The work [218] seems to confirm this theory: 52 children aged 5 to 7 (including 26 TD children and 26 children with ASD) took part in the experiment. Two images of faces were presented to them on a screen, and the children’s task was to indicate their preferred face. Each child made 90 choices by pressing a button, and eye movements were recorded with an eye tracker. The order of the image pairs was randomized, and their left–right order was counterbalanced. Nine female faces were used: three cartoon faces, three human faces, and three intermediate faces that were the result of transforming the cartoon face into a human face using Abrosoft FantaMorph software. The results confirm the presence of the uncanny valley phenomenon in TD children. In contrast, it is not observed in children with ASD: their preference remains the same as face realism increases. However, the absence of the uncanny valley in these children cannot be explained by inattention to the elements being manipulated: the eye movement recordings indicate that the entire area concerned was scanned, and the time spent looking into the eyes of the faces is similar in both groups. Nevertheless, the gaze patterns diverged, with the autistic children staring less at the pupils and more at the area between the two eyes and the area peripheral to the pupils. This latter aspect is consistent with the literature indicating an atypical gaze pattern in individuals with ASD [58]. In addition, an autistic child does not show an increase in heart rate when confronted with a robotic face considered frightening by a TD child [219], which can also be explained by a lack of uncanny valley.

How can we explain the absence of this phenomenon in individuals with ASD? It is important to note that neurotypical infants under 12 months of age do not demonstrate the uncanny valley phenomenon [220]. The formation of the face prototype would thus contribute to the emergence of this phenomenon. However, people with ASD have a deficit in face prototype formation [221,222]. More specifically, children with ASD have limited visual experience of human faces. One reason for this is their low motivation to orient to social stimuli [152]. As a result, they are less sensitive to facial features such as eye size or changes in the distance between the eyes [223]. Furthermore, in the general population, the fewer autistic traits individuals have, the easier it is for them to differentiate a human-controlled robot from one acting autonomously [224]. The ability to finely detect an agent’s peculiarities could therefore be impacted in autistic individuals, and it is these subtle cues that would identify the agent’s divergence from the norm. Due to reduced sensitivity to subtle changes in facial features, autistic individuals would be less inclined to perceive this discrepancy and would therefore present a reduced (or even absent) uncanny valley phenomenon [218]. TD individuals thus show an aversion to android robots (strongly resembling humans) and prefer those with a weakly or moderately human appearance [213], while autistic individuals would not show this aversion [218,219]. We then ask whether they explicitly show a preference for one type of robot over another when asked directly about it.

A study conducted with 16 adolescents with ASD aged 10 to 17 (IQ ≥ 85) compared their preferences for different robots of more or less human-like appearance [172]. The experimenter presented three humanoid robots: an android robot (ACTROID-F), an iconic robot (Smile Supplement), and a mechanical robot (M3-SYNCHY) in three separate sessions lasting an average of 5 min. The participant’s task was to answer three questions: “Do you like robots?”, “What was your favorite robot?”, and then “What was your second favorite robot?”. All the participants indicated that they liked robots, but there were significant differences in preferences between the three robot types. It is generally considered that individuals with ASD prefer a robotic appearance rather than a human one [86,203], so the authors had postulated that the preference of autistic individuals would be for the more mechanical-looking robot (M3-SYNCHY). However, half the participants expressed a preference for the Smile Supplement robot, which resembles a human-shaped plush toy: the iconic robot (Smile Supplement) was identified as the favorite by eight participants for its cuteness and tenderness, four participants preferred the mechanical robot (M3-SYNCHY) for its machine-like appearance, and four participants preferred the android robot (ACTROID-F) for its advanced technology [172]. The simplest (least human-like) robot is therefore not necessarily the one preferred by autistic adolescents. On the other hand, contrary to all expectations, the individuals with the most severe autistic traits reported a preference for the android robot (the most human-like). However, in a preliminary study, many children under the age of eight were afraid of the android robot because of its realism [172]. The authors conclude that, as long as the stimulus is simplified, it is of interest to individuals with ASD. In other words, although android robots are not the most simplified stimuli (through their strong resemblance to a human, in terms of appearance—and sometimes behavior), they remain simpler than humans, allowing autistic individuals to be less overwhelmed [85]. Furthermore, knowing that individuals with ASD do not seem to be sensitive to the uncanny valley due to their limited visual experience of human faces, they do not necessarily reject robots that strongly resemble humans, although this may vary according to age and degree of disorder severity. These results highlight the variability in preferences among autistic individuals, which constitutes a considerable challenge in the use of robots with children with ASD.

### 5.2. Interindividual Variability

An individual-level analysis reveals significant variations in reaction when interacting with robots [202]. Some autistic children interact with robots in the same way as they would with an object, while others treat them as social agents [108]. The literature also points to considerable variability in preference for particular robots [104] depending on the child’s characteristics and in particular the type of interaction he or she uses [172]. These heterogeneous results underline the importance of taking into account the idiosyncrasy of these disorders and symptomatology in these differences in interaction and preference. The same robot may not be suitable for all children with ASD as the differences in abilities and needs that exist between two individuals are very marked [60]. Not all children with ASD benefit equally from the presence of a robot [41].

User-related factors such as cognitive (IQ) or verbal abilities, sensory preferences, gender, age, and culture can influence the quality of the interaction [85] and consequently the benefits provided by the robot.

ASD can be associated with extremely different cognitive functioning, ranging from profound intellectual disability to high potential [225].More precisely, high-functioning autism corresponds to a mild degree of ASD, with an IQ in the average range (IQ ≥ 70). Conversely, individuals with low-functioning autism are more severely affected by ASD, with an IQ < 70 [226], and require greater support. The impact of the participant’s cognitive functioning on interaction with the robot remains to be clarified. Indeed, one study found more curiosity in low-functioning children than in high-functioning ones, with no other differences in activity engagement [227]. Yet, autistic children with profound intellectual disability (IQ = 22) did not benefit from imitation-based interaction with an NAO robot, unlike children with severe to mild intellectual disabilities [4,52,228]. The authors explain that these children were unable to complete the task, which underlines the importance of personalizing interactions, with the development of specific approaches to supporting individuals with profound intellectual disabilities. The benefit may also depend on the skill targeted: an autistic child with a low IQ (IQ = 52, which corresponds to moderate intellectual disability) does not progress in verbal communication following interaction with a robot, but they do show an improvement in their emotional response similar to that of other children with a higher IQ [201]. IQ differences would thus have an impact on interaction efficiency but also on preferences: children with a high IQ (IQ > 100) prefer android robots, unlike children with a lower IQ [172]. However, many studies have focused on high-functioning children with autism (see, for example, [146,147]). The results obtained in the literature cannot therefore be generalized to the entire autistic population insofar as an intellectual disability is associated in approximately 30% of the cases [229].

Individuals with ASD also show marked differences in language skills: some individuals are non-verbal, while others have preserved expressive language, albeit with difficulties in the pragmatic aspect of language [60]. Non-verbal children are less likely to engage in interactions with a human or robot than pre-verbal children (children who communicate through gestures, vocalizations, and looks but not words) or verbal children [31,227].

Children’s sensory preferences also have an impact on their interaction with a robot in a joint attention or imitation task. Children with ASD who are hypo-sensitive to visual stimuli have more difficulty following the robot’s instructions [230] and imitating it [231]. Individuals with ASD reporting high levels of sensory sensitivity found it easier to interact with an android robot when it made few movements. Conversely, participants with lower levels of sensory sensitivity found it easier to interact with a robot with many movements [232].

Participants’ gender could modulate his or her perception of robots. In TD children, the results are contradictory [233,234]. In a study of children with ASD, boys associated robots more with machines than girls for four of the six robots studied (PLEO, NAO, KEEPON, and ROMIBO) [104]. Nevertheless, these results were gathered from a small sample size, so further studies are needed to establish a potential effect of gender on the perception of robots in the autistic population. Given that women with ASD show better social interaction and communication skills than men with ASD in human–human contexts, notably in social attention [235] (see [236] for a review) and social cognition [237], this difference may also be observed in human–robot interactions starting in childhood: TD girls show more social motivation than boys [238]. Nevertheless, to our knowledge, no effect of gender on the interaction of autistic children with a robot has yet been demonstrated.

A meta-analysis suggests that the benefits of robots for autistic individuals would be more marked when the child is young [17], particularly in the area of verbal communication [198]. The link between optimal age and robot type remains undetermined, but it seems that choices evolve with age. Younger users would prefer a small, simple-looking robot, while teenagers make other choices [85]. In one study, the oldest participants (aged 17) preferred an android robot, while the younger children (aged around 10) preferred a plush or mechanical robot [172]. Children under 8 are even said to be afraid of android robots.

Finally, the benefits of robots could vary according to culture: Japanese autistic children would show more engagement in interaction with a robot than Serbian children [199]. In contrast, another study highlighted differences between Japanese and French children when interacting with a human but not when interacting with a robot [99]. While Japanese children show a significantly higher heart rate than French children when interacting with a human, this is not the case when interacting with a robot, and French and Japanese children report the same emotional state. Furthermore, the significant improvement in emotional state between the beginning and end of interaction with the robot was similar for both groups.

In general, autistic children with the fewest autistic symptoms, the best language skills, and the best social functioning initiate more interactions with a robot and with an adult [60] and derive more benefit from interacting with a robot [165]. Two main consequences of ASD variability can be highlighted [60]: we must remain vigilant about over-generalizations given the heterogeneity of the autistic population; this also implies that individualized, even personalized, interventions for each child where possible could better match their needs. A specific type of interaction may be better suited to one child than another depending on the child’s symptom profile. The individual scale makes it possible to use the child’s specific interests [239] and individual characteristics to adjust the difficulty of the interaction content [19,240] as well as the robot’s appearance [241]. Personalizing interaction sessions according to the child’s preferences and abilities thus enables increased engagement [202]. Activities that are familiar to the child lead to greater engagement by the child compared with unfamiliar activities [47]. For these reasons, focusing on a sub-group of children sharing certain characteristics could facilitate the design of interventions and their effectiveness [60] although the ideal still remains to create each interaction according to individuals’ needs. For example, a robotic intervention for children with high support needs (with severe symptoms, and often limited language skills) would be based on simple speech and basic social skills training. Conversely, this type of intervention would not be suitable for an autistic child with higher language skills, who would require more elaborate speech. Thus, the interactions of autistic children with a robot should be personalized to match the needs of each individual. Various features of these interactions can be adapted: the complexity of the robot’s behaviors (its speech and movements), the type of task proposed (which may be familiar or new, of low or high difficulty, and related or not to the child’s specific interests), and even the robot’s appearance. Indeed, although we have observed that individuals with ASD seem less sensitive to the uncanny valley phenomenon, they may show a preference for one appearance over another, which needs to be taken into account in the design of these interactions. Finally, the choice in robot type will also determine the software used for teleoperation. It is therefore preferable to favor robots that allow flexible and accessible programming (e.g., the NAO robot with Choregraphe software) so that interaction customization can be carried out successfully.

The differences observed between children with ASD highlight the importance of identifying the individual characteristics of each child, which are essential if the results are to be generalized. For the time being, however, this rigor is rarely put into practice [242]. This makes it difficult to generalize the results. In addition, methodological limitations prevent us from reaching solid conclusions.

### 5.3. Methodological Limitations

In the course of this review, we have noted that the methodology employed is highly heterogeneous, both in the choice of robot used and in the type of intervention proposed, which vary according to the objective pursued (see Table 1 and Table 2 for a summary). Several methodological limitations have been highlighted in studies involving the interaction of autistic children with a robot. Firstly, few studies compare the progress made by interacting with a robot with that made by interacting with a human, which does not allow demonstrating the robot’s superior effectiveness. Yet, this control condition is necessary to reach a solid conclusion on the benefits provided by a robot. The presence of a comparative group of TD children would also be relevant to better define the differences between ASD and TD children when interacting with a robot. Secondly, some authors point to the small sample size (which would concern more than half of the studies), as well as the low participation of girls, absent in 10% of the studies [60,205,243]. Furthermore, in many studies, sample characteristics are poorly described: mean age and standard deviation are not systematically available, nor are participants’ cognitive abilities. Some samples include a wide age range (for example, one study included participants aged from 5 to 24 [49]). Yet, children’s gender, age, and culture, as well as their cognitive, language, and sensory profile, can have an impact on their interaction with a robot. The characteristics of the participants therefore need to be detailed as far as possible, and then taken into account in the intervention and in the interpretation of the results. The lack of standardized criteria for assessing the functioning of people with ASD [60,205] and robot acceptance [244] also constitutes a limitation in the study of interactions with robots. Nevertheless, we can note that the most recent studies are improving the rigor of their methodology on the points cited above.

Finally, it remains to be determined (1) how these benefits evolve throughout the intervention; (2) whether they are maintained over time once the intervention is over, and for how long; and (3) whether they generalize to interactions with humans in an ecological context. One problem is the lack of long-term follow-up of the sessions: interventions generally consist of one or two sessions, whereas a minimum of ten is required. With a single session, it is difficult to differentiate the effectiveness of the robot from the novelty effect, which could bias the results [242]. In addition, these studies may conclude that the robot is effective in the short term, but they do not guarantee that these benefits will be maintained over time [243,245]. However, long-term exposure appears to be beneficial. While some studies show a decrease in eye contact with the robot throughout the interaction [29,180], others highlight that the improvement in behavior is maintained over several sessions [47,246] or even increases as children show more and more engagement in the activity as time goes on [26,202]. This benefit would be all the greater for children with the most severe autistic symptoms, compared with peers with a mild to moderate form of autism [202]. In addition, the skills acquired during training with a robot would be maintained after the end of the intervention: after a delay of 2 weeks [20,65,70], 4 weeks [13], and even up to 3 months [52]. Robots could also help to maintain progress over time. A study that included an assessment 3 months after the end of the intervention showed that training with a robot generated greater benefits than training with a human [113]. The same question can be asked about the generalization of these behaviors to human partners once the sessions with the robot are over: the abilities acquired with the robot then seem to be transferable to human–human interactions [19,21,38,52,65]. The children would be able to apply the skills acquired with the robot in new contexts, at least for joint attention [39,40], gesture imitation [65], and emotion recognition [21]. Parents of children with ASD who underwent a month of training with the robot around social games reported significant changes in their child’s social behavior: increased eye contact, interaction initiation, and response to communicative acts [19]. This point remains to be clarified, however, as it is rarely evaluated. Long-term studies (e.g., [246]) are therefore of crucial interest in evaluating the benefits brought by robots to people with ASD, to specify the evolution of these benefits during and after the intervention, as well as their transfer to new interaction contexts.

Generally speaking, the exploratory nature of most studies and the lack of methodological rigor prevent us from drawing any solid conclusions regarding the benefits of using a robot with the autistic population [243]. Nevertheless, we can note that the most recent studies tend to propose a more rigorous methodology with regard to the above-mentioned points. A recent meta-analysis seems to confirm the benefits of robot use on the development of social skills, but not on emotional and motor skills [17]. Further studies are therefore needed to establish the usefulness of social assistance robots for individuals with ASD indisputably [17,83]. Future studies should include standardized measurement tools and systematically specify the sample selection process and sample characteristics. Because of the significant interindividual variability among individuals with ASD, robot-based interventions should be personalized according to the child’s characteristics (cognitive and language abilities, sensory profile, and specific interests).

## 6. Conclusions

In conclusion, the use of robots with people with ASD appears to be beneficial in encouraging communication and social interaction, supporting the learning of specific social behaviors, and reducing maladaptive behaviors. Interaction with robots could improve the social skills of individuals with ASD [17]. Children with autism touch and look more at a robot than at a human [15,28,30]. They also show a greater tendency to follow a robot’s gaze than a human’s gaze [43]. Children’s participation in an activity is increased when a robot is present [45,47], thus increasing interaction initiation [14,46] and speech production [44]. During an action imitation task performed by a human or robotic arm, children with autism perform better with the robotic arm, while TD children perform better with the human arm [53]. Furthermore, when ToM training is implemented, children with ASD make more progress with a humanoid robot than with a human [11], and the same results are observable in emotion recognition training with an iconic robot [21]. Finally, studies suggest that the stress felt by individuals with ASD during social interactions could be reduced with a robotic partner [50], leading to a decrease in the occurrence of stereotyped behaviors [28,45]. Further studies are needed to confirm the influence of robots on stereotyped behavior, but the benefits of robots on social development have been confirmed by a recent meta-analysis [17].

Although autistic children may categorize robots in the same way as TD children, they seem more attracted to robots and attribute more human characteristics to them. Some studies even suggest that theyare more interested in robots than in humans.

We sought to explain the benefits of robots by drawing on two theories: Social Motivation Theory and Social Cognition Theory. On the one hand, considering Social Motivation Theory, individuals with ASD may show more interest in a non-human agent than in a human one. This increased interest in interaction would then allow them to accumulate social experience, which could consequently reduce their social difficulties. On the other hand, as Social Cognition Theory argues that people with ASD engage less in interactions due to difficulties in understanding the social world, it is conceivable that a simpler, more predictable agent such as a robot could reduce these difficulties. As a result, this type of agent would encourage autistic individuals to engage more fully in social interactions. The robot, as a more attractive, simplified, and predictable social agent than a human, would thus encourage social interactions in autistic individuals. For children with significant social difficulties, robot-assisted training could thus constitute a step of intermediate difficulty compared to training with a human [182].

It therefore seems appropriate to rely on this type of agent when assessing the social skills of individuals with ASD: by reducing the difficulty of the interaction, a robot could enable children to better mobilize their abilities, leading to better estimation of their social skills. This suggests that psychology tests might be more successful if the experimenter is a robot rather than a human, as has already been shown in TD children [247,248,249,250,251]. The robot would then provide a more accurate means of assessing social–cognitive functioning [252,253], which would be particularly relevant for children with ASD [254].

However, there are still several points to be clarified in the study of interactions between autistic individuals and robots. Firstly, the ideal type of robot does not yet seem to have been identified. The preferences of autistic children diverge from those of neurotypical children since they show no aversion to robots that strongly resemble humans. This distinction could be explained by a lower sensitivity to the physical irregularities of robots [172]. However, among children with autism, preferences may vary from a robot that bears little resemblance to a human to one that bears a strong resemblance. Secondly, the impact of individual characteristics (cognitive and language abilities, sensory profile, age, gender, and culture) on the outcome of interaction with a robot is yet to be defined [198]. This aspect is all the more important as ASD is characterized by wide interindividual variability [1]. Thirdly, studies often have methodological limitations: small sample sizes and lack of precision on sample characteristics, low participation of girls, lack of control groups, and lack of long-term follow-up sessions. Nevertheless, it seems that recent studies are tending towards greater methodological rigor, improving each of these aspects.

In summary, it seems that individuals with ASD can benefit from interaction with robots, but questions remain and will need to be resolved in the future (not least because of interindividual variability and methodological limitations). In particular, we may wonder whether robots benefit only a sub-group of individuals with ASD or whether their use is relevant to the entire population. It will also be necessary to examine whether the benefits observed during interaction with a robot can be maintained over time, and whether the abilities fostered by the robot can be transferred to an environment other than the clinical or experimental context. Some results indeed indicate a generalization of these skills in human–human interactions in the absence of the robot [21,38,52,65].

For future studies, the recommendations are to (1) take individual factors into account to set up a personalized interaction tailored to the child’s needs; (2) develop a rigorous methodology (in particular with a control condition and a large sample whose characteristics are detailed); and (3) design long-term interactions to examine the evolution over time of the benefits brought by a robot and the generalization of the skills acquired. Finally, given the benefits of the robot on communication and social interaction, it may be relevant in the future to include a robot in assessments of the social functioning of children with ASD. In this way, social skills could be measured in a simplified interaction context, enabling autistic children to better mobilize their skills, which would then be more accurately assessed.

## Figures and Tables

**Table 1 behavsci-14-00131-t001:** Benefits of robots in fostering communication and social interaction in individuals with ASD.

Section	Article	Variable	Effect	Effect *p*-Value	Robot	Sample Size (ASD)	Mean Age (Standard Deviation)	Functioning/Mean IQ (Standard Deviation)	Duration of Robot Intervention (mn)	Country
Eye contact	Barnes et al. [24]	Eye gaze	robot > human	NA	NAO	3	8.33 (4.04)	NA	1 session (NA)	USA
	Bekele et al. [25]	Eye gaze	robot > human	p<0.05	NAO	6	2.78–4.9 y.o.	NA	1 session (30–50 mn)	USA
	Cao et al. [26]	Eye gaze	robot > human	p<0.05	NAO	15	4.96 (1.10)	NA	1 session (NA)	China
	Costa et al. [27]	Eye contact	robot > human	p<0.001	KASPAR	8	6–10 y.o.	NA	7 sessions (NA)	UK
	Costa et al. [28]	Eye gaze	robot > human	p<0.05	QTRobot	15	9.73 (3.38)	IQ < 80 (*n* = 8); 80–120 (*n* = 6); IQ > 120 (*n* = 1)	1 session (1.5–4.3 mn)	Luxembourg
	Damm et al. [29]	Eye gaze	robot > human	p<0.05	FLOBI	9	21 (NA)	112.5 (range: 94–133)	NA	Germany
	David et al. [30]	Eye contact	robot > human for 2/5 children	p=0.01	NAO	5	3–5 y.o.	LF-HF	8–12 sessions (10 mn/session, 1/day)	Romania
	Duquette et al. [31]	Eye contact	robot > human	β=0.35	TITO	4	5.1 (NA)	LF	3/week for 7 weeks (NA)	Canada
	Scassellati et al. [19]	Eye contact with other people	pretest < post-test	p<0.01; p<0.05	JIBO	12	9.02 (1.41)	IQ ≥ 70	23 sessions (30 mn/session, 1/day)	USA
	Shamsuddin et al. [32]	Eye contact	robot > human	NA	NAO	1	10 y.o.	107	1 session (15 mn/session)	Malaysia
	Simlesa et al. [15]	Eye contact	robot > human	p<0.05	NAO	12	5.2 (0.63)	MA: 2–3 y.o.	1 session (NA)	Croatia
	Simut et al. [33]	Eye contact	robot > human	p<0.05	PROBO	30	6.67 (0.92)	91.23 (range: 70–119)	1 session (15 mn/session)	Belgium
	Tapus et al. [34]	Eye gaze	robot > human	p<0.05	NAO	4	4.2 (1.67)	NA	7–13 sessions (8 mn/session, 2/day)	Romania
	Wainer et al. [35]	Eye contact	robot > human	p<0.05	KASPAR	6	6–8 y.o.	NA	2 sessions (15 mn/session)	UK
Joint attention	Anzalone et al. [36]	Joint attention	robot < human	p<0.01	NAO	16	9.25 (1.87)	73 (14)	1 session (NA)	France
	Cao et al. [26]	Joint attention	robot = human	p>0.05	NAO	15	4.96 (1.10)	NA	1 session (NA)	China
	Cao et al. [37]	Joint attention	robot < human	p<0.001	NAO	27	46.37 months (4.36)	MA: 42 months max	2 session (NA)	The Netherlands
	Ghiglino et al. [14]	Joint attention	robot = human	p>0.05	Cozmo	24	5.79 (1.02)	58.08 (19.39)	5 weeks (10 mn/session)	Italy
	Kajopoulos et al. [38]	Joint attention	pretest < post-test	p<0.05	CuDDler	7	4–5 y.o.	NA	6 sessions over 3 weeks (20 mn/session)	Singapore
	Kumazaki et al. [39]	Joint attention	time +; robot > human	p<0.01	CommU	28	70.56 months (6.09) (robot group); 69.00 (4.39) (control)	NA	1 session (15 mn)	Japan
	So et al. [40]	Joint attention	pretest < post-test (robot)	p<0.05	HUMANE	38	7.51 (0.87) (robot group); 7.91 (0.89) (human group)	LF	6 sessions (30 mn/session)	China
	Taheri et al. [41]	Joint attention	time +	p<0.05	NAO/ALICE-R50	6	6–15 y.o.	LF-HF	12 sessions over 3 months (30 mn/session)	Iran
	Warren et al. [42]	Joint attention	time +	p<0.01	NAO	6	3.46 (0.73)	NA	4 sessions over 2 weeks (NA)	USA
	Wiese et al. [43]	Gaze cueing effect	robot > human	p<0.01	EDDIE	18	19.67 (1.5)	NA	1 session (15 mn)	Germany
Interaction	Ghiglino et al. [14]	Social interaction initiation	robot > human	p<0.05	Cozmo	24	5.79 (1.02)	58.08 (19.39)	5 weeks (10 mn/session)	Italy
	Kim et al. [44]	Social behaviors towards peer	robot > human	p<0.05	PLEO	24	4–12 y.o.	NA	NA	USA
	Pop et al. [45]	Collaborative game	robot > human	p<0.05	PROBO	11	4–7 y.o.	IQ >70	8 sessions (1 mn/session)	Romania
	Pliasa et al. [46]	Social interaction initiation	robot > human	p<0.05	DAISY	6	6–9 y.o.	NA	2 sessions (20 mn/session)	Bulgaria
	Rakhymbayeva et al. [47]	Engagement time	tendency time 2 > time 1; familiar > unfamiliar activities	p=0.05; p<0.05	NAO	7	6.1 (2.7)	LF	7–10 sessions (15 mn/session)	Khazakstan
	Stanton et al. [48]	Social interaction initiation	robot > human	p<0.05	AIBO	11	5–8 y.o.	NA	1 session (30 mn)	USA
	Wainer et al. [35]	Cooperation in game	robot > human	p<0.01	KASPAR	6	6–8 y.o.	NA	2 sessions (15 mn/session)	UK
Touch	Costa et al. [27]	Spontaneous touch	robot > human	NA	KASPAR	8	6–10 y.o.	NA	7 sessions (NA)	UK
	Simlesa et al. [15]	Touch	robot > human	p<0.05	NAO	12	5.2 (0.63)	MA: 2–3 y.o.	1 session (NA)	Croatia
Communication	Farhan et al. [49]	Verbal and non-verbal communication	time 4 > time 1	NA	NAO	4	5, 12, 13, 24 y.o.	range: 41–47	4 sessions (NA)	Bangladesh
	Huskens et al. [13]	Self initiated questions	pretest < post-test; robot = human	p<0.05; p>0.05	NAO	6	3-14 y.o.	IQ >80	4 sessions (10 mn/session)	Netherlands
	Kim et al. [44]	Speech	robot > human	p<0.05	PLEO	24	4–12 y.o.	NA	NA	USA
	Kumazaki et al. [50]	Posture, gaze, facial expressions	robot > human	p<0.001	ACTROID-F	29	29.1 (2.6)	IQ ≥ 70	1 session (25 mn)	Japan
	Simlesa et al. [15]	Vocalization	robot < human	p<0.01	NAO	12	5.2 (0.63)	MA: 2–3 y.o.	1 session (NA)	Croatia
	Stanton et al. [48]	Speech	robot > human	p<0.05	AIBO	11	5–8 y.o.	NA	1 session (30 mn)	USA
	Syrdal et al. [51]	Communication	number of interactions +	p<0.05	KASPAR	19	2–6 y.o.	NA	NA	UK
	Taheri et al. [41]	Verbal communication	time +	p<0.05	NAO/ALICE-R50	6	6–15 y.o.	LF-HF	12 sessions over 3 months (30 mn/session)	Iran
Imitation	Conti et al. [52]	Imitation	time +	NA	NAO	6	5 and 10 y.o.	LF	15 sessions (6–8 mn/session)	Italy
	Costa et al. [28]	Imitation	robot = human	p>0.05	QTRobot	15	9.73 (3.38)	IQ < 80 (*n* = 8); 80–120 (*n* = 6); IQ > 120 (*n* = 1)	1 session (1.5–4.3 mn)	Luxembourg
	Duquette et al. [31]	Imitation of facial expressions; of words and gestures	robot > human; robot < human	NA	TITO	4	5.1 (NA)	LF	3/week for 7 weeks (NA)	Canada
	Pierno et al. [53]	Imitation velocity	robot > human	p<0.001	ROBOTIC ARM	12	11.1 (NA)	HF	1 session (NA)	Italy
	Simlesa et al. [15]	Imitation	robot = human	p>0.05	NAO	12	5.2 (0.63)	MA: 2–3 y.o.	1 session (NA)	Croatia
	Soares et al. [54]	Imitation of emotions	robot > human; post-test > pretest (robot), post-test = pretest (human)	p<0.05; p<0.05; p>0.05	Zeno	45	6.8 (1.5) (robot group); 7.5 (1.4) (human group); 7.8 (1.2) (control)	HF	2/week for 3 weeks (5–15 mn/session)	Portugal
	Taheri et al. [55]	Imitation	robot < human	p<0.001	NAO	20	4.95 (2.01)	NA	1 session (NA)	Iran
	Zheng et al. [56]	Imitation quality	robot > human	p<0.05	NAO	6	3.83 (0.54)	NA	2 sessions (NA)	USA

HF: High-Functioning Autism; IQ: Intellectual Quotient; LF: Low-Functioning Autism; MA: Mental Age; NA: Not Available; y.o.: Years Old.

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
