# Peer review of "People with Autism Spectrum Disorder Could Interact More Easily with a Robot than with a Human: Reasons and Limits"

_behavsci, 2024, doi:10.3390/bs14020131_

Round 1

Reviewer 1 Report

Comments and Suggestions for Authors

The overall level of this paper is very good, and it is well written, but please consider the following issues:

1)    In the introduction, the paper lacks to explain how the 49 papers were selected (exclusion and inclusion criteria). It is clear that the papers were selected to answer research questions (Q1-Q4). Please clearly explain the selection process. If possible, add methodology section please, that would be better.

2)    In the discussion that starts at Line 938: please include more (Section 5) about (usability, user acceptance, and accessibility of socially assistive robots) and how the robots are supposed to be designed to suit users’ needs and disabilities and according to the targeted user group. In some studies, the (uncanny valley) phenomenon is referred to as a usability barrier.  The challenges here should mainly be associated with the way the interfaces (software and hardware interfaces) were designed, getting benefit from the studies that highlight how the deficits of autistic persons affect the interaction in order to design better robotic interfaces for them. The limits are made by the way we design not by autistic people's deficits.

Reviewer 2 Report

Comments and Suggestions for Authors

Review by authors has divided it by types of behaviors but not necessarily by age (EG https://pubmed.ncbi.nlm.nih.gov/35731805/). There are similar reviews as recent as 2022. Can authors take this information and extrapolate whether these behaviors have sustained effect on social interactions life or do they limit what this population may want to achieve EG Do we know if this population desires to have a bias towards robots given the robot itself may be a reflection of their facial expressions or contricted affect.

All in all the different aspects of behaviors is a strength of this article. Using training by robots for this population can be highlighted to help this population move towards gainful employment. 

Possibly highlight Robotoc tool versus an actual robot (or a ful size human like robot). What is the clinical significance of interactions and who does it benefit and in what specific way can be highlighted to emphasize robotic-neurodivergent population.

Reviewer 3 Report

Comments and Suggestions for Authors

This work presents a large review of articles on human-robot interactions in the case of people with autism. It is of definite interest if only for this review. However, it seems to me that it would benefit from being revised on certain points. 

In particular, the question of 'social interaction' as such is not addressed, even though it lies at the heart of both this article and the references cited. I think it's problematic that we never mention, or at least devote a section to, what we mean by 'social interaction'. This would also help to flesh out your conclusion, as well as a critical point concerning your table (Table 1). Visualising a set of studies as you present it masks, or rather ignores, methodological disparities (which you mention elsewhere, but too superficially in my opinion), and there is a smoothing effect. Somewhere you sketch out the notion of the 'robot as a social agent', which is an important subject that deserves a little more attention.

 Critics like the one in section 2.1 L141-144 is the kind of interesting « step back » i’m talking about.

I suggest to look at references that address the topic of « social interaction with robot » or « engagemennt » per se :

Dickerson  P, Robins  B and Dautenhahn  K (2013) Where the action is : A conversation analytic perspective on interaction between a humanoid robot, a co-present adult and a child with an ASD. Interaction Studies 14(2): 297-316.

Rollet N and Clavel C (2020) “Talk to You Later”: Doing Social Robotics with Conversation Analysis. Towards the Development of an Automatic System for the Prediction of Disengagement. Interaction Studies 21(2): 269-293.

Comments on some sections : 

3.2.

can you explain what is the difference for you between « preference » and « increased interest » ? Is it normative wise ?

3.2. I would suggest here to mention some references about what categorization is, and I suggest to be very ervy carefull addressing this topoic for categorization process is quite different whether it emerges during an interaction (say during a human-robot interaction), or if it is a discursive device used in a questionnaire.

A lead : 

Alač M (2016) Social robots: Things or agents? AI & Society 31(4): 519-535.

Castañeda C and Suchman L (2014) Robot visions. Social Studies of Science 44(3): 315-341.

Rollet N and Clavel C (2020) “Talk to You Later”: Doing Social Robotics with Conversation Analysis. Towards the Development of an Automatic System for the Prediction of Disengagement. Interaction Studies 21(2): 269-293.

Rollet N, Jain V, Licoppe C, et al. (2017) Towards Interactional Symbiosis: Epistemic Balance and Co-presence in a Quantified Self Experiment. In: Gamberini L, Spagnolli A, Jacucci G, et al. (eds) Symbiotic Interaction: 5th International Workshop, Symbiotic 2016, Padua, Italy, September 29–30, 2016, Revised Selected Papers. Cham: Springer International Publishing, pp.143-154.

+ the work of Ethnomethodology and Conversation Analysis on categorisation processes

4.1

I don’t find any explanation of why ASD chidren would me more « motivated », more oriented towards social interaction when encountering robots. You just say that robot is a motivator, but why is that so.

There is another problem to me : you assume that human-robot exchange is a social interaction, which needs to be demonstrated. I seems to me that due to the purpose of your demonstration you assume a homogeneous category « robot-as-a-social-agent » which is absolutely not obvious to me.

4.2.. Do you suggest that ASD children may be more incline to interact with robots because they are more predictable systems than humans ? 

+ How would one explains the fact that triangular interaction ASD-chidren-therapist-Robot, seems beneficial, whereas it appears to be a more complex interactional system than human-human or human-robot ? Or maybe not ?

6.

interesting conclusion, but when you write « further studies are required to confirm that the use of robots benefits all inidivudals with autism », you , in a way, say to your reader that all the effort made in your bibliographical work coud have been avoided, summarized in a single Table, and then object of constructive discussion, instead of a very long list of examples of studies as you did. I mean, you conclusion relies on the nuance supported by the  adjective « all » (individuals) -L1083.

Thank you for the reading !
